# Aftershocks are fluid-driven and decay rates controlled by permeability dynamics

Stephen A. Miller [1✉]

One aspect of earthquake physics not adequately addressed is why some earthquakes generate thousands of aftershocks while other earthquakes generate few, if any, aftershocks. It also remains unknown why aftershock rates decay as ~1/time. Here, I show that these two are linked, with a dearth of aftershocks reflecting the absence of high-pressure fluid sources at depth, while rich and long-lasting aftershock sequences reflect tapping high-pressure fluid reservoirs that drive aftershock sequences. Using a physical model that captures the dominant aspects of permeability dynamics in the crust, I show that the model generates superior fits to observations than widely used empirical fits such as the Omori-Utsu Law, and find a functional relationship between aftershock decay rates and the tectonic ability to heal the co- and post-seismically generated fracture networks. These results have far-reaching implications, and can help interpret other observations such as seismic velocity recovery, attenuation, and migration.

[1] Center for Hydrogeology and Geothermics (CHYN), University of Neuchâtel, Neuchâtel, Switzerland. ✉email: stephen.miller@unine.ch

Aftershock sequences display a wide spectrum of behavior (Supplementary Table 1). For example, a great earthquake (Mw = 8) in Peru in 2019 and a major earthquake (Mw = 7.1) in Mexico in 2017 generated no aftershocks detectable by the global seismic network (M ≥ 4). The M7.7 Jamaica earthquake (2020) immediately generated a cluster of 18 aftershocks (M ≥ 4) at the western boundary of the main rupture (likely via stress transfer), and followed by only five additional aftershocks. By contrast, the M7.1 Ridgecrest, CA. earthquake (2019) generated over 130,000 aftershocks[1] in the first weeks, including over 100 M ≥ 4 events. Eight foreshocks (M ≥ 4.5 and M ≤ 6.5) preceded the M6.4 Puerto Rico earthquake (2020), spawning 85 aftershocks (M ≥ 4) in the subsequent three weeks. One can argue that the intermediate depths of the Peru (78 km) and Mexico (48 km) earthquakes, or the oceanic crust of Jamaica, are mixing apples with oranges when comparing to the shallow crustal Ridgecrest and Puerto Rico earthquakes, but even directly comparing apples raises questions. For example, the 2001 Kunlun (M7.8) earthquake was a surface-rupturing crustal strike-slip earthquake with a lateral extent of about 400 km. The 2002 Denali (M7.9) earthquake was also surface-rupturing crustal strike-slip earthquake with a lateral extent of about 400 km. The Kunlun earthquake generated a total of 12 aftershocks (M ≥ 4) in the first three weeks, with eight of those occurring in the first 24 h. This differs (by more than an order of magnitude) with the Denali earthquake, which was preceded by a M6.1 foreshock, and generated 157 aftershocks (M ≥ 4) in the subsequent three weeks, 90 of which occurred in the first 24 h.

One plausible explanation for these large differences in aftershock behavior is the availability of deeply trapped high-pressure fluids to drive the aftershock sequence. In the cases of Peru and Mexico (Supplementary Fig. 1), geodynamical and petrological arguments, combined with observations, suggest that these normal faulting events (at the transition from flat to steep subduction) occurred in-between two different dehydration horizons, and thus relatively fluid-absent. The Jamaica earthquake ruptured an oceanic transform fault, which is known to generate relatively few aftershocks[2], but it was a major earthquake that generated a paucity of aftershocks relative to its size (and with no obvious deep fluid sources in the oceanic lithosphere). In contrast, Puerto Rico is integrated into the Caribbean subduction system, with potentially abundant deep fluid sources available, while Ridgecrest is underway in and near the Coso hydrothermal region with significant deep $CO_2$ and $H_2O$ sources capable of driving this sequence. Finally, an obvious candidate for the order-of-magnitude discrepancy observed for the Kunlun and Denali earthquakes is that Tibet rests upon 65–80 km of crust[3] thickened by the Himalayan orogeny, with no obvious mechanism for generating deeply derived fluids, while an active subduction zone (with the concomitant abundance of volatiles) lies 60 km beneath Denali[4]. These observations suggest that, in the absence of trapped and high-pressure fluid reservoirs at depth, aftershocks are suppressed, while access to deep fluid sources produce robust and long-lived aftershock sequences.

The prevailing view of aftershocks is that they result from a combination of stress-triggering by changes in Coulomb Failure Stress (ΔCFS)[5,6], changes in stressing rates[7] that influence rate-state friction, or as a response to viscous relaxation processes[8]. Although widely used, ΔCFS models often find unsatisfactory correlations between aftershock locations and stress changes associated with the mainshock, particularly since most aftershocks occur in regions adjacent to the main rupture plane in the stress-relieved regions after the main-shock. Immediate off-fault aftershocks, such as the Landers-triggered Big Bear (1992) event, are likely triggered by dynamic or Coulomb stress changes. Statistical models such as ETAS[9] nicely fit aftershock data because

each aftershock is allowed to produce its own sequence, but currently lacks any physical basis. An alternative view is that aftershocks are fluid-driven from pore pressure diffusion[10–12], or at least that high fluid pressures are involved[13–19]. Since rocks fail by some combination of increasing Coulomb failure stress or increasing pore fluid pressure, the dominant mechanism is likely the one with the widest range of amplitudes. ΔCFS models that seem to correlate, if they correlate[5], are on the order of fractions of MPa, while expected changes in fluid over-pressure can range from zero to the minimum principal stress (e.g., hundreds of MPa at 10 km depth).

The underlying physical mechanism driving aftershocks of large earthquakes has remained elusive since Omori[20] first empirically observed in 1894 that aftershocks decay as approximately 1/t. The modified Omori-Utsu Law[21] (among many of its variations[22]) is an empirical fit to the number of aftershocks with time $n(t)$ following an earthquake:

$$n(t) = \frac{A}{(c+t)^p} \qquad (1)$$

where $t$ is time and $A$, $c$, and $p$ are constants used to fit the data. In this work, I normalize by the maximum number of aftershocks, so the productivity parameter A is 1. To date, the only physical models that predict Omori-type behavior are the rate-state formulation of friction[23] and damage models[24].

Here I propose a hypothesis that long-duration aftershock sequences are predominantly fluid-driven, and test the hypothesis by comparing results of a physically-based non-linear diffusion numerical model against observations of tens of thousands of aftershocks following three large earthquakes in California. I show that this (physical) model produces superior fits to aftershock rates than widely used empirical laws, and find strong spatial correlations with measured hypocenters and the calculated elevated fluid pressure for all sequences investigated. I also show that aftershock rates are controlled by the orientation of the cluster with respect to the direction of the maximum horizontal stress (SHmax) because SHmax represents the dominant mechanism for permeability recovery.

## Results

### Study area and data used for comparisons with model results.
Figure 1a shows all well-located aftershocks[25] (M ≥ 1.5) generated for the 1992 (M7.3) Landers, the 1999 (M7.1) Hector Mine, and Fig. 1b for the 1994 (M6.7) Northridge earthquakes. The Landers and Hector Mine earthquakes were right-lateral strike-slip faults that generated tens of thousands of aftershocks, and the Northridge earthquake ruptured a blind thrust fault and also produced a robust, but temporally aborted, aftershock sequence.

Also shown in Fig. 1 are some A-quality stress measurements of the orientation of SHmax determined from the World Stress Map[26]. These data are relevant because the azimuth of SHmax, relative to the orientation of the aftershock cluster, is shown below to contribute to controlling the aftershock decay rate. Each of the curves (Fig. 1c) is modeled in this work. Figure 1c shows the first 1000 days of aftershocks (normalized) for each of the named sequences modeled in this study.

### A model for permeability dynamics.
A conceptual model of permeability dynamics constrains a formulation for permeability that mimics the essence of permeability dynamics through the earthquake cycle[27]. Permeability dynamics is simplified to three dominant processes; (1) an exponential dependence of permeability on the effective normal stress[13,28], (2) a step-wise increase in permeability at the onset of slip[29], followed by; (3) an exponential decrease in permeability with time mimicking post-

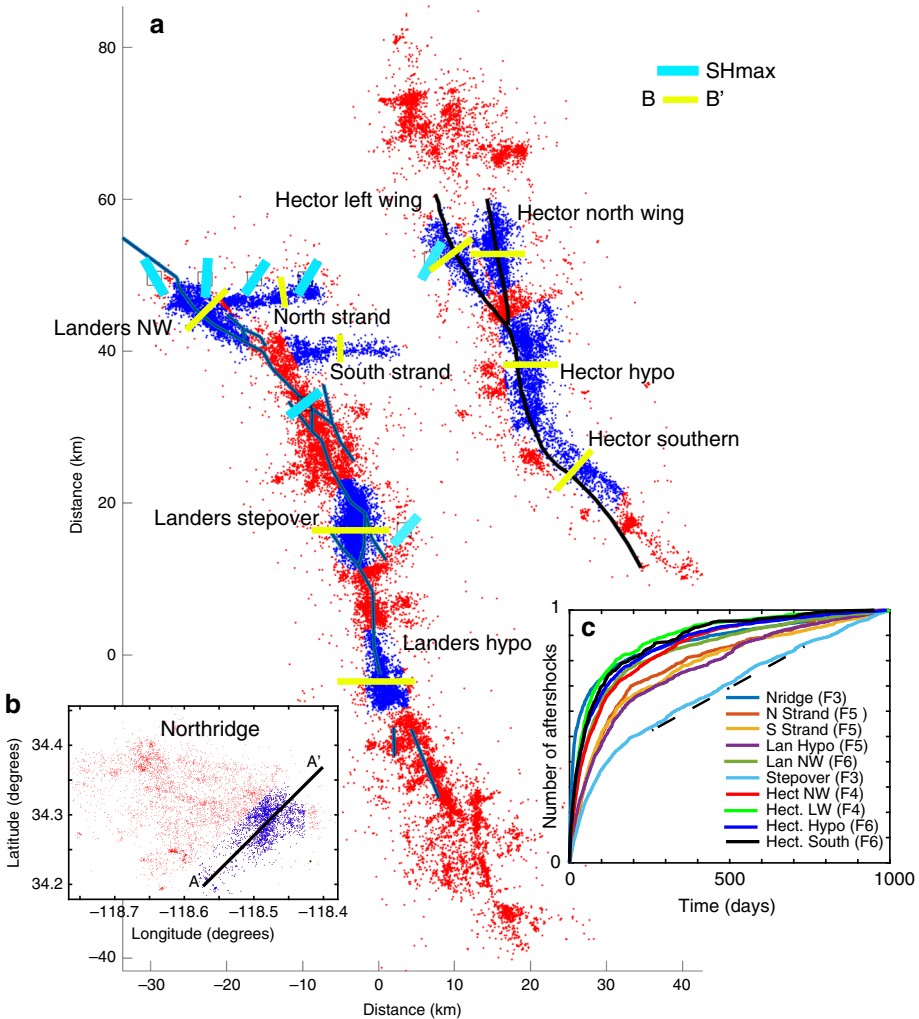

**Fig. 1 Study area and data.** Map (**a**) showing epicenters of all M ≥ 1.5 aftershocks (red) from the 1992 Landers (M7.3), the 1999 Hector Mine (M7.1), and (**b**) the 1994 Northridge (M6.7) earthquakes. Blue dots show the datasets used to compare with model results. For Northridge, aftershocks were projected onto profile A-A' for depth distributions, and all other profiles B-B' are shown in yellow. Light blue lines show orientations of the maximum horizontal stress (SHmax) obtained from the World Stress Map[26]. Names are given to each cluster for easier reference in the text. (**c**) The cumulative number of aftershocks (normalized by the maximum number in each catalog) for all of the aftershock sequences modeled in this study. The legend identifies the Figure number for comparisons between model results and observations. The dashed line marks the onset of a constant aftershock rate for the Stepover.

seismic healing/sealing of the permeability network. Therefore, permeability prior to failure is expressed as[28]:

$$k = k_o \exp\left[-\sigma_e/\sigma^*\right]; \tau < \mu\sigma_e \quad (2)$$

where k is permeability [m²], $\sigma_e$ is the effective normal stress [Pa] (e.g., $\sigma_e = \sigma_n - P$), τ is the shear stress [Pa], μ is the friction coefficient, P is pore pressure [Pa], and $\sigma_*$ [Pa] constrains the permeability response to $\sigma_e$.

Permeability increases co-seismically because the creation of a fracture or slip on a frictional surface may increase permeability by orders of magnitude[29], while permeability recovery via tectonic stresses and/or crack healing and sealing is assumed to decrease exponentially in time. This results in the following formulation for permeability after failure:

$$k = (k_o + d\exp(-\alpha t))\exp\left[-\sigma_e/\sigma^*\right]; \tau > \mu\sigma_e \quad (3)$$

where d is a co-seismic step-increase in permeability at slip [m²] to mimic the increase in permeability associated with a fresh fracture or slip on a pre-existing frictional interface. Permeability

then exponentially decreases over a timescale defined by α [s⁻¹], returning to its effective-stress-dependence at long times.

Substituting Eq. (3) into the 2D diffusion equation results in a non-linear diffusion model that is solved using Finite Differences (Methods).

**Model constraints and initial conditions.** Figure 2 shows the numerical setup for the thrust fault case (Fig. 2a) and all of the strike-slip simulations (Fig. 2b) where a zone of high pore pressure (assumed as 70 MPa above hydrostatic) is imposed within a domain of background permeability. This initial setup approximates the accruing evidence that high slip velocities concomitant with earthquake rupture raise temperatures sufficiently to either expand and pressurize existing pore fluids, called thermal pressurization[30,31], or via direct generation of high-pressure fluid sources through temperature-induced dehydration and decarbonatization. Theoretical studies of thermal pressurization focus on the expansion of entrained pore fluids in thin fault zones, and show that fluid pressures quickly diffuse because of the very limited fluid volume involved. However, dehydration and decarbonation reactions[32] may also provide direct (and

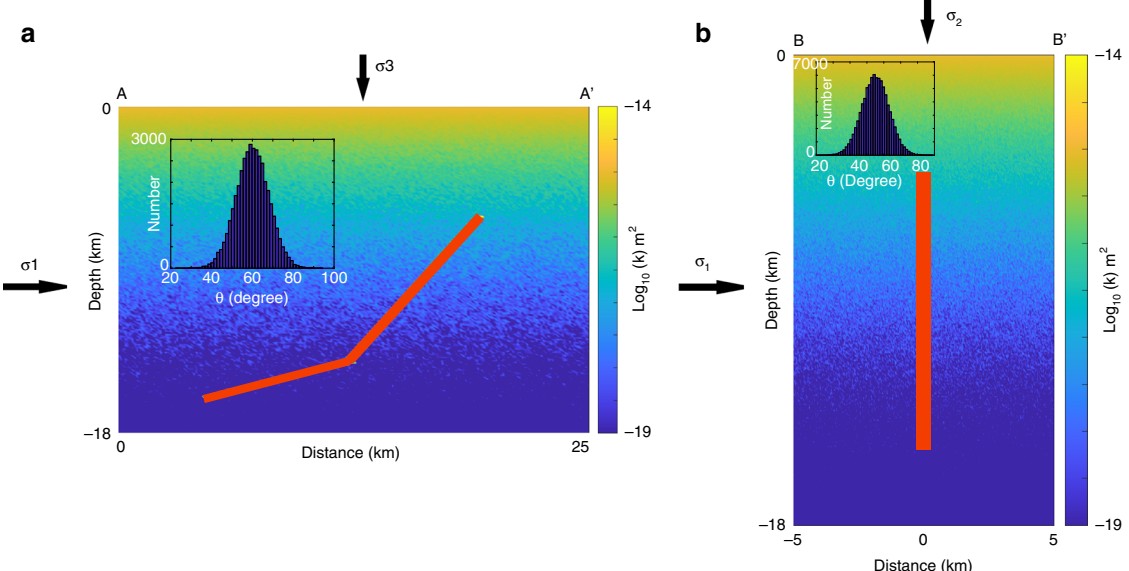

**Fig. 2 Model setup and initial conditions.** Model setup for (**a**) Northridge and (**b**) all Landers and Hector Mine simulations, showing an initial overpressure embedded within the shown background permeability field. The positioning of the two-fault Northridge simulation was determined by the cloud of aftershocks in cross-section, and all other simulations were centered in the middle of the profile. The red indicates the initial zone of overpressure, and set at 70 MPa above hydrostatic. Inset histogram shows the orientation of virtual slip planes measured from the normal to $\sigma_1$ (taken as twice $\sigma_3 = \rho gz$), where $\rho$ is rock density, $g$ is gravity acceleration, and $z$ is depth.

substantial) high-pressure fluid sources within the fault zone that then diffuse into co-seismically generated damage zones[33] at timescales that depend on the permeability structure. Recent numerical studies[34] demonstrated a new and additional mechanism for generating high fluid pressures co-seismically by self-pressurization of fluid-bearing fault zones through elasto-plastic collapse of rapidly deforming fluid-filled pores. The co-seismically generated fault zone also provides a highly permeable conduit that can link to deeply trapped, supra-hydrostatic fluid reservoirs[18,35]. Since the geometry and location of trapped high-pressure fluid reservoirs are not known, I do not include them in this modeling study. Conceptually, an over-pressured fault zone would only enhance fluid flow because it would be highly permeable (Eq. 3 at $t = 0$), and thus an open channel for draining the hypothesized reservoirs at depth.

The parameters controlling model behavior (Eq. 3) are $k_0$, $d$, $\sigma^*$, and $\alpha$. The values for $k_0$ and $\sigma^*$ were chosen to result in an initial permeability field (Fig. 2a, b) that roughly approximates models of crustal permeability[36], and $d$ was chosen as 1000, a reasonable estimate for crack-induced permeability enhancement. This is discussed in more detail below.

**Comparisons between model results and observations**. Model results are compared with observations for each cluster (blue dots in Fig. 1) by comparing hypocentral depths with the calculated pressure field (above hydrostatic), and the rate of numerical aftershocks with rates of observed aftershocks (inset Fig. 1). A numerical aftershock is defined as a numerical gridpoint reaching the Mohr-Coulomb failure condition, and its time and location are recorded (Methods and Supplementary Fig. 2). The ZMAP[37] statistical bootstrapping algorithm was used to determine the best-fitting $p$-value and $c$-value of the Omori-Utsu Law. ZMAP fits to the data were determined for the same time window as that for model comparisons to the data.

Northridge and the Landers Stepover: The best way to demonstrate the versatility of the model (Eq. 3) is to compare two end-member cases of Northridge and the Landers Stepover.

Northridge, a thrust earthquake, provided an efficient mechanism (through large horizontal compressional stresses) to heal the co-seismically generated fracture network, implying fast permeability recovery (i.e., a high value for $\alpha$). The Landers Stepover is a right-lateral step in a right-lateral fault system (kinematic pull-apart) with no obvious mechanism for tectonically healing the co- and post-seismically generated fracture networks, thus implying sluggish permeability recovery (i.e., a low value for $\alpha$). Model results (Fig. 3d, h) are consistent with this view, where good correlations are found between model and observations with $\alpha = 7.5 \times 10^{-3}$ days$^{-1}$ (i.e., $1/\alpha = 133$ days) for Northridge and $\alpha = 4 \times 10^{-4}$ days$^{-1}$ (i.e., $1/\alpha = 2500$ days) for the Landers Stepover. Importantly, this is reflected in the $p$-value of Eq. (1), where $p = 1.28$ for Northridge and $p = 0.66$ for the Stepover (Fig. 3c, g), a first indication that the p-value is tectonic in origin. Model comparisons of aftershock rates with the Stepover data are limited to 300 days because of the observed onset of a constant aftershock rate (Fig. 1). A constant rate of aftershocks cannot currently be achieved with this model, but may indicate that substantial deep fluid sources continued to recharge this region, particularly because kinematic pull-aparts represent efficient channels for rapid deep degassing and mineral deposition[38]. This will be addressed in future studies.

Strong spatial correlations are found when comparing thousands of hypocentral depths and the fluid pressure field (Fig. 3b, f) calculated through the permeability network (Fig. 3a, e) evolved from the initial permeability fields (Fig. 2). Note that the good comparison of computed pressure fields and data are shown as snapshots in time, but the model cannot match the data at early times because some events occur outside the diffusion front (Supplementary Fig. 3). Identifying a diffusion front in the data requires much more extensive coverage than existed for these earthquakes, and migration has already been identified in the extensively covered Ridgecrest earthquake[1].

The calculated record of the average permeability (Fig. 3i) shows permeability for Northridge begins rapid recovery after about 200 days, while no recovery is observed for the Stepover case and the system remains highly permeable for the duration

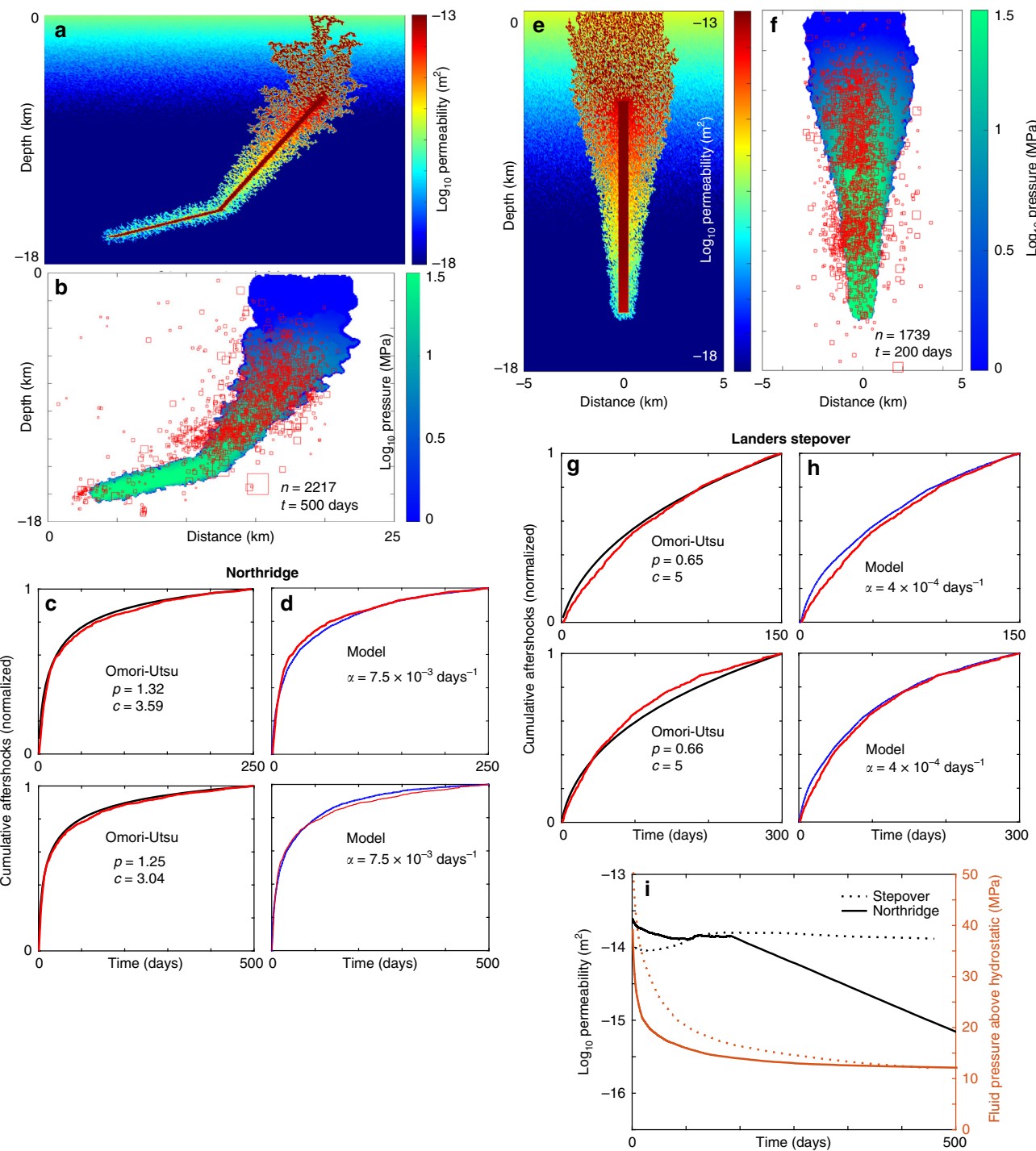

**Fig. 3 Model results for Northridge and the Stepover.** For Northridge, (**a**) the permeability network evolved from the initial background permeability field (Fig. 2a). (**b**) Comparison of 1722 measured hypocenters and the calculated fluid pressure field calculated at 500 days. (**c**) comparison of data (red) and the Omori fit (black) using the ZMAP algorithm for two different time periods. (**d**) same data (red) compared to model results (blue) for the same time periods. For the Stepover (**e**) the permeability network evolved from the initial background permeability field (Fig. 2b). (**f**) Comparison of 1739 measured hypocenters and the calculated fluid pressure field calculated at 200 days. (**g**) comparison of data (red) and the Omori fit (black) at two different time periods. (**h**) same data (red) compared to model results (blue) for the same time periods. (i) Average pressure and permeability time histories of the stimulated regions for both cases, showing rapid permeability recovery for Northridge and negligible recovery for the Stepover.

simulated. The calculated average fluid pressure within the evolved network remains elevated and stabilizes to about 10 MPa as diffusion slows and large initial pressure gradients smooth out. An observation that may be related to over-pressures trapped over longer timescales is the observation that the Hector Mine earthquake triggered a swarm of activity in the step-over region of

Landers (Supplementary Fig. 4). Although Hector Mine would not necessarily be considered distant-triggering at Landers, triggering of earthquake clusters is almost exclusively limited to hydrothermal regions (c.f. Figure 8, ref. [39], which would have been the state of the Landers step-over if trapped high-pressure fluids remained.

It is important to note that a good match to the data was achieved for the Landers Stepover even though $\alpha$ played no role over the timescale of the simulation. This indicates that both $\sigma^*$ and $\alpha$ (Eq. 3) contribute to the behavior of this model, and laboratory experiments are needed to better quantify these parameters. Achieving a good match also indicates that the values are chosen for $\sigma^*$ and $d$ (Methods) are reasonable for the strike-slip cases, and were therefore used for all subsequent simulations. This should be emphasized; all simulations presented below were identical to the Stepover case, with the exception of the value for $\alpha$.

Hector North Wing and Hector Left Wing: The observations and modeling results for Northridge and the Stepover provides a working hypothesis. Namely, higher p-values indicate faster permeability recovery than relatively lower p-values. A good test of this hypothesis is to investigate two separate aftershock clusters spawned by the 1999 Hector Mine (M7.1) earthquake, one cluster trending North-South (called North Wing in Fig. 1), and one trending roughly NW-SE (called Left Wing in Fig. 1). Quality A

stress measurements are limited in this area, but there is one measurement near the Hector Mine earthquake that indicates an azimuth of SHmax at 30° (light blue bar in Fig. 1), roughly consistent with other stress measurements in the region. An azimuth of 30° implies that the optimal orientation for slip is on faults striking N-S and E-W. This optimal orientation for slip also provides the highest permeable pathway because of the lowest effective normal stress (first term of Eq. 2). Thus, the hypothesis predicts that the Left Wing recovers faster than the North Wing because the maximum tectonic horizontal compressive stress approaches the normal to the aftershock trend, while the N-S strike of the North Wing experiences the lowest fault-normal compression needed to heal the co- and post-seismically generated fracture network.

To model this, I simply re-ran the Landers Stepover case but used $\alpha = 3 \times 10^{-3}$ days$^{-1}$ (i.e., $1/\alpha = 333$ days) for the North Wing and $\alpha = 6 \times 10^{-3}$ days$^{-1}$ (e.g., $1/\alpha = 167$ days) for the Left Wing. Figure 4 compares observations with modeling results for these two systems, and a few points should be noted. The first is

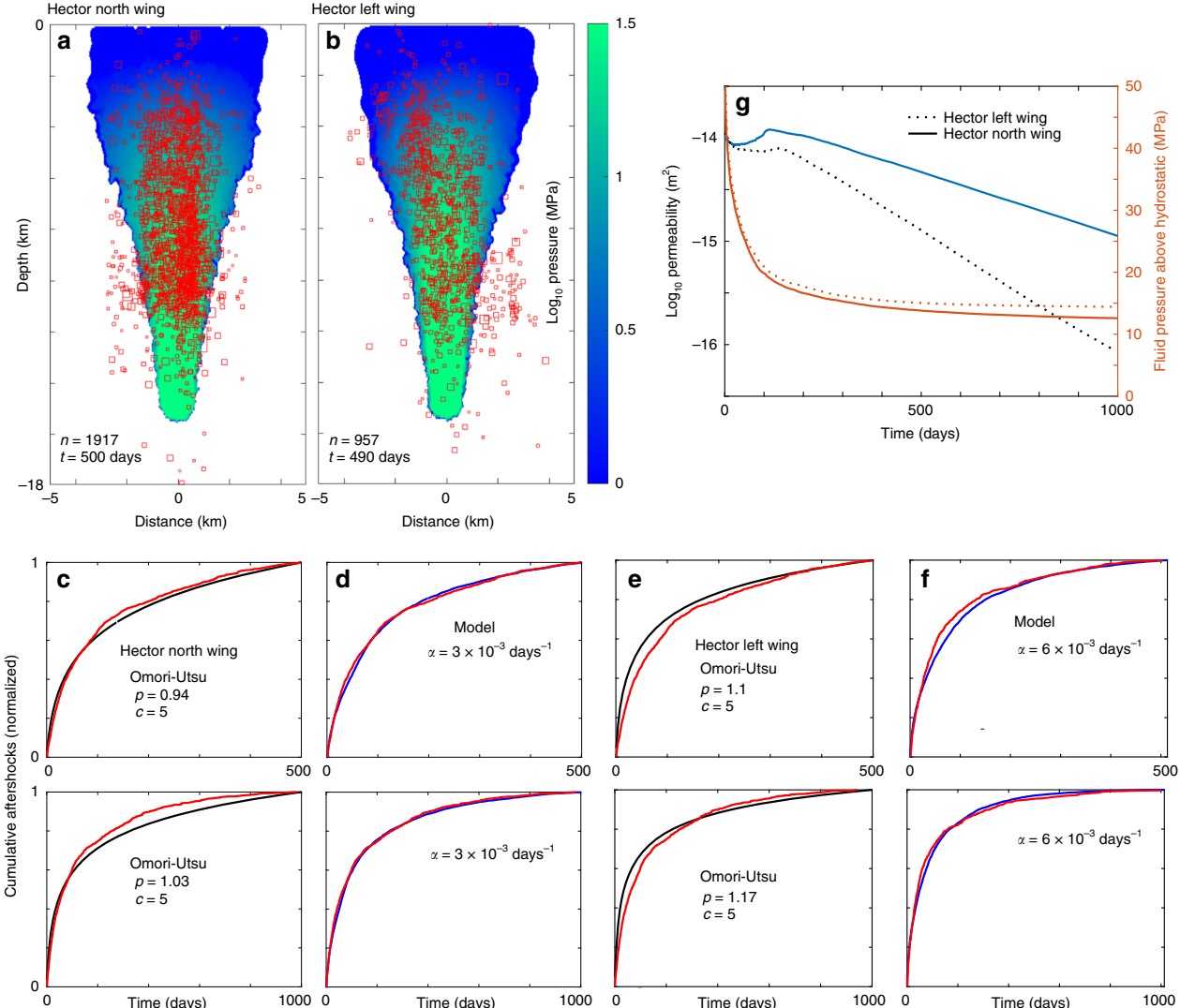

**Fig. 4 Model results for Hector North Wing and Left Wing.** Comparison of 1917 measured hypocenters and the computed fluid pressure field for (**a**) North Wing and (**b**) Left Wing (957 hypocenters). (**c**) comparison of data (red) and the Omori fit (black) for two different time windows, and (**d**) same data (red) compared to model results (blue) for the same time windows for North Wing. (**e**) comparison of data (red) and the Omori fit (black) for two different time windows, and (**f**) same data for Left Wing (red) compared to model results (blue) for the same time periods. (**g**) Average pressure and permeability time histories of the stimulated region showing Left Wing recovers permeability significantly faster than North Wing. Note the different p-values.

that the best-fitting *p-value* of the empirical Omori-Utsu Law (Eq. 1) is time-window dependent (Fig. 4c, e), with different p-values for different time-windows. Second, the *c* value of Eq. (1) that best fits the data is 5 days, even though statistical and seismological studies constrain *c* to be on the order of hours[40], if not less. Any fit using a *c* value close to zero is poor in any time-window, rendering Eq. (1) inadequate even as an empirical fit. This observation is not widely known because Omori-Utsu is traditionally investigated in logarithmic space, which obscures badly fitting data. By contrast, the modeled and observed cumulative aftershocks for both sequences (Fig. 4d, f) show excellent (and time-window independent) agreement, providing additional support that the *p-value* is an indicator of permeability recovery. The final and most important point about the aftershock rates for these two systems is that $p \sim 0.95$ for the North Wing compared to $p \sim 1.1$ for the Left Wing case, as the hypothesis predicts. That is, the NW-SE trending sequence recovers permeability quicker than the N-S trending sequence because fault-normal compression heals the fracture network.

The comparison of measured hypocenters and calculated fluid pressure for both cases (Fig. 4a, b) show that a vast majority of the data is encased by the region of calculated fluid-overpressure, albeit with some outliers in both cases. It is also intriguing that the productivity of the North Wing is more than twice that of the Left Wing, consistent with the North Wing being more amenable to fluid flow. The time-histories of average permeability and fluid-overpressure (Figures g) quantify these values, showing that after a few years, permeability recovers enough to limit diffusion processes, and importantly because of this, overpressures of about 10 MPa are left trapped in the crust, to diffuse over much longer timescales.

Landers Hypocenter, North Strand, and South Strand: These three seemingly disparate sections of the Landers earthquake all share the common feature that they are in dilatant sections of the main fault rupture, and their time histories (Fig. 1) show very similar behavior. With similar time-histories, they can be modeled with similar values for *α*. The p-values for these three sequences are approximately 0.7 (Supplementary Fig. 5), indicating little permeability recovery. Figure 5 shows modeling results

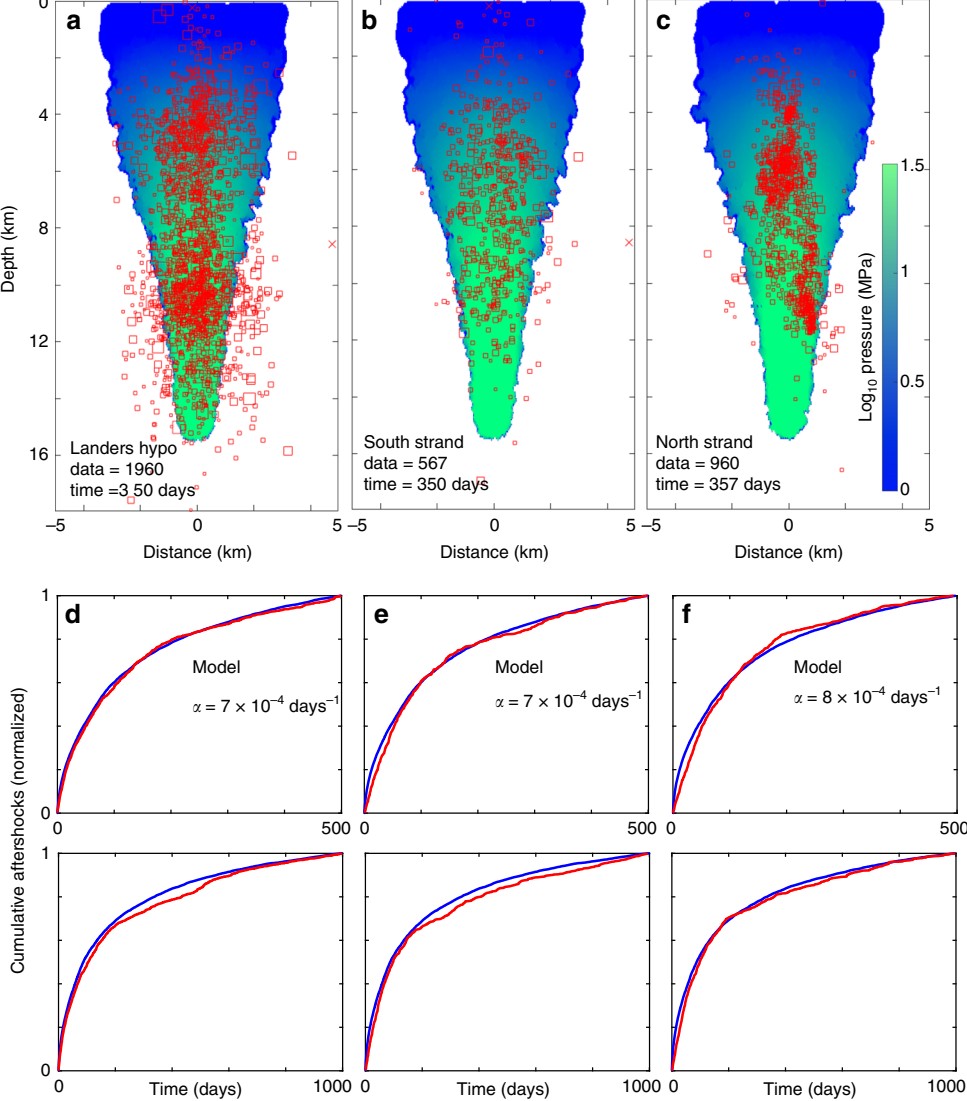

**Fig. 5 Model results from Landers Hypo, the South Strand, and the North strand. (a–c)** Comparison of computed pressure fields with "data = " measured hypocenters for three different dilatant sections of the Landers earthquake. (**d–f**, **e**) Comparison of data (red) and model results (blue) for two different time windows, showing that despite dramatic differences in productivity, all three sections exhibit very similar permeability recovery times. See Supplementary Fig. 5 for the corresponding Omori fits.

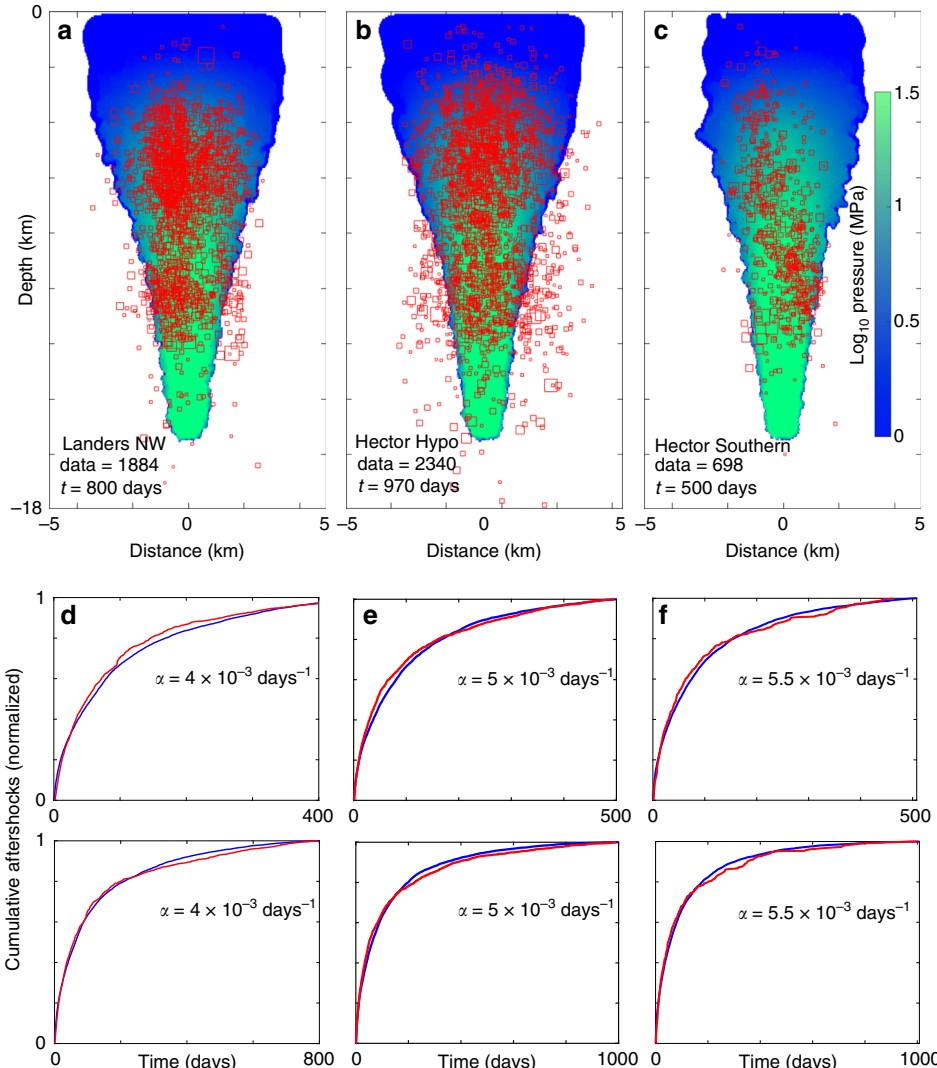

**Fig. 6 Model results from Landers NW, Hector Hypo and Hector Southern. (a–c)** Comparison of computed pressure fields with "data = " measured hypocenters for three different dilatant sections of the Landers and Hector Mine earthquakes. (**d–f, e**) Comparison of data (red) and model results (blue) for two different time windows, demonstrating again that dramatic differences in productivity results in similar permeability recovery times. See Supplementary Fig. 6 for the corresponding Omori fits.

using $\alpha = 7 \times 10^{-4}$ days$^{-1}$ for Landers Hypo and South Strand, and $\alpha = 8 \times 10^{-4}$ days$^{-1}$ for the North Strand, and demonstrates that, again, the calculated fluid pressure encases a vast majority of observed hypocenters (Fig. 5a–c), and that the aftershock rates are well-matched by the model (Fig. 5d–f).

Landers NW, Hector Hypo, and Hector Southern: The final three simulations in this study (Fig. 6) show very large differences in productivity, but similar permeability recovery rates defined by $\alpha$. As with all cases studied, the calculated elevated pressure field (Fig. 6c) encases most of the observed aftershock hypocenters Fig. 6a–c).

A-quality stress measurements near the northwest extent of the Landers earthquake (Landers NW) indicates that this sequence occurred along a fault system optimally-oriented for slip and thus low normal stress (e.g., high permeability), and consequently an expected behavior similar Hector's North Wing (Fig. 4). The ZMAP fit to the data (Supplementary Fig. 6) shows a time-window dependence with $p$ ranging of 0.9 to 1.03, very similar to that observed for the Hector North Wing, and an overall fair to poor fit in all time windows. Simulating this involved changing $\alpha$ from $3 \times 10^{-3}$ days$^{-1}$ of the North Wing case to $4 \times 10^{-3}$ days$^{-1}$ for the Landers Northwest case. The comparison between

observed and numerical aftershocks (Fig. 6d) shows excellent agreement in all time windows, and that the calculated pressure field (Fig. 6a) encases almost all observed aftershocks.

Figure 1 (inset) suggests that the Hector Hypo dataset falls somewhere in between the North Wing and Left-Wing cases (e.g., $\alpha_{\text{RightWing}} < \alpha_{\text{Hypo}} < \alpha_{\text{LeftWing}}$). With this in mind, I changed $\alpha$ to $5 \times 10^{-3}$ days$^{-1}$, resulting in similarly good fits with the data (Fig. 6e). A vast majority of aftershock hypocenters are encased within the elevated fluid pressure field (Fig. 6b), albeit with more outliers than other simulations.

Extrapolating the orientation of SHmax to the southern section of the Hector Mine earthquake (Hector Southern) implies that this sequence healed relatively quickly because the orientation of the cluster is almost normal to the azimuth of the maximum stress direction. Indeed, modeling using $\alpha = 5.5 \times 10^{-3}$ days$^{-1}$ shows excellent temporal correlation with observations (Fig. 6f) compared to the ZMAP fit to the data (Supplementary Fig. 6) that shows a persistent time-window dependence and an overall fair to poor fit to the data in all time windows.

Modeling other aftershock sequences from the three earthquakes studied here becomes redundant because the spatial

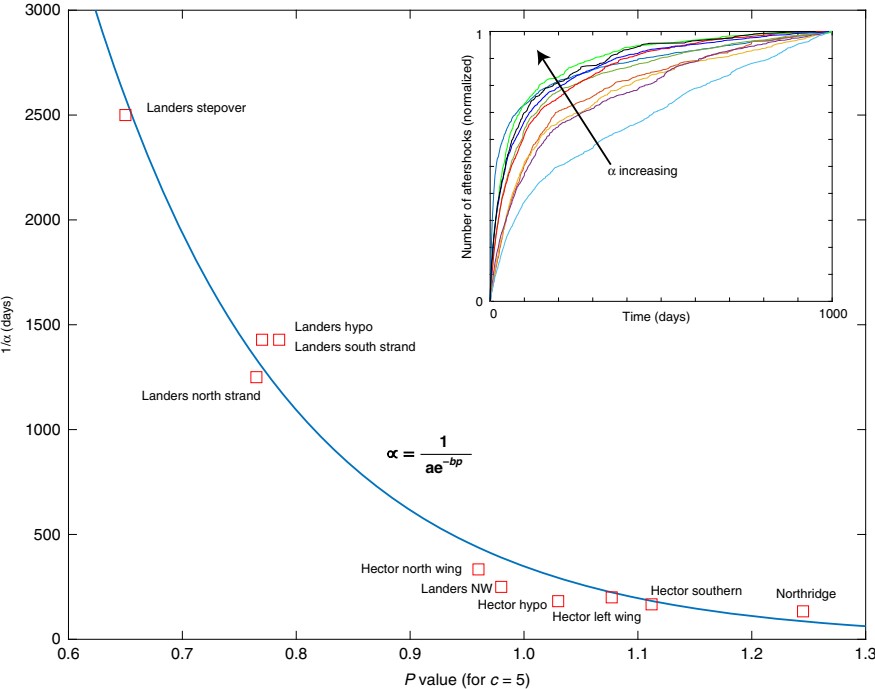

**Fig. 7 Synthesis of modeling results. Data reveals an exponential relationship (95% confidence.** bounds) between the Omori-Utsu p-value and the timescale of permeability recovery. Inset shows all of the time- histories modeled in this study (see Fig. 1 for names of datasets), and shows that the shape of the curve is a measure of the permeability recovery time (α).

distribution of hypocenters in the x-z plane always looks roughly the same, the model results always look the same, and the only difference between each sequence is the temporal decay of aftershocks, which in the model is controlled by α.

A compendium of calculated pressure and permeability histories (Supplementary Fig. 7) quantifies permeability and pressure evolution for each of the simulations presented for comparison between all simulations.

**Synthesis of results**. Figure 7 synthesizes the modeling results, where I plot the ZMAP-determined p-value of Omori-Utsu against 1/α with units [Days] for intuitive convenience. The p-value plotted is the average of the p-values determined for each time window. The results show an exponential relationship between the p-value and 1/α, indicating that the shape of the aftershock rate (inset, and Fig. 1) is a direct measure of permeability recovery, and which can be used as a lookup table for assessing permeability recovery of past and future earthquakes.

# Discussion

Using a conceptually simple and physically plausible model for permeability dynamics in the crust, I numerically demonstrate that aftershock decay rates reflect the tectonic ability to re-seal the co- and post-seismically generated fracture networks. Fracture healing through time suppresses subsequent fluid flow and thus aftershock production. Long-duration and rich aftershock sequences are expected in extensional pull-aparts and zones of dilatation because these regions have little tectonic ability to shut down the fracture networks, while compression-dominated events such as Northridge produce truncated aftershock sequences because compressional stresses easily close co-seismically generated opening for propagating deeply trapped fluids. That mega-thrust earthquakes typically generate long-lasting aftershock sequences might reflect the continuous generation of fluids through dehydration during subsequent large

aftershocks, or draining of trapped and high-pressure fluids likely associated with episodic tremor and slow-slip earthquakes[41–43]. Regions that are geodynamically dry will produce few, if any, aftershocks. These results have far-reaching implications, and can be used to interpret other observations such as seismic velocity recovery[44], attenuation[45] and migration[1,46].

# Methods

**Non-linear Diffusion Model**. The finite difference numerical model solves the diffusion equation;

$$\frac{\partial P}{\partial t} = \frac{1}{\phi \beta} \nabla \cdot \frac{k}{\eta} \nabla P$$

where P is fluid pressure above hydrostatic, $\phi$ is porosity (taken as .03), $\beta$ is the lumped compressibility (taken as 1e-8 $Pa^{-1}$) of the pore space and fluid[47], $k$ is the permeability, and $\eta$ is the dynamic viscosity of the fluid (taken as $10^{-3}$ at the surface and decreases with depth). Permeability $k$ prior to failure takes the form[28]:

$$k = k_o \exp\left[-\sigma_e/\sigma^*\right]; \tau < \mu\sigma_e$$

where $k_o$ is taken as 1e-15 $m^2$ when a grid point is below the failure condition, $\sigma_e$ is the effective normal stress (e.g., normal stress minus pore pressure), and $\sigma^*$ (taken as 35 MPa) defines how permeability changes in response to $\sigma_e$. When a numerical grid point reaches the Mohr-Coulomb failure condition permeability changes in time via:

$$k = (k_o + d \exp(-\alpha t)) \exp\left[-\sigma_e/\sigma^*\right]; \tau > \mu\sigma_e$$

where permeability undergoes a stepwise increase d (taken as $10^{-12}$ $m^2$ for Landers, and Hector Mine, and $10^{-11}$ $m^2$ for Northridge) to mimic the increase in permeability associated with slip or fracture[29]. Such high values of permeability have been inferred for some aftershock sequences[48]. Permeability then exponentially decreases (e.g., heals the network) over a timescale defined by α [plotted for convenience in days$^{-1}$], returning to its effective-stress-dependence at long times.

The model calculates pore pressure diffusion through an evolving permeability field (e.g., Fig. 3a, e) that takes a step-wise increase when a numerical grid point reaches the Mohr-Coulomb failure condition (Supplementary Fig. 2).

**Simplified Mechanics**. Mechanics are only virtually simulated by assigning a (normally-distributed) angle θ to each numerical grid point, where θ is the angle measured from the normal to $\sigma_1$. Effective normal stress $\sigma_e$ and shear stress $\tau$ are

resolved on these planes via:

$$\sigma_e = \frac{\sigma_1 + \sigma_3 - 2P}{2} + \frac{\sigma_1 - \sigma_3}{2}\cos(2\theta)$$

$$\tau = \frac{\sigma_1 - \sigma_3}{2}\sin(2\theta)$$

In the strike-slip case, the virtual slip planes are out of the page, explained in detail elsewhere[49]. Since θ affects the effective normal stress (and thus permeability), the initial stress and permeability states are heterogeneous. As high fluid pressure diffuses into the hydrostatically-pressured surroundings, the stress state of a numerical grid point reaches the failure condition and a numerical aftershock is counted and located at that time. Further details of this model can be found elsewhere[49] because it is the same model[49] calibrated against the meso-scale fluid-injection experiment in Basel (Switzerland).

For the thrust fault cases, $\sigma_1$ was assumed as twice $\sigma_3$, which was taken as the weight of the overburden, and the fluid pressure $P$ (above hydrostatic) was calculated. The initial overpressure was assumed as 70 MPa above hydrostatic to along a 450 m wide fault zone. Such a wide fault zone is likely overestimated, but does not have any significant influence on the results. The model domain was 300 × 300 finite-difference nodes, and no flow boundary conditions were imposed on the bottom and sides of the model, and a zero-pressure boundary at the top surface.

## Data availability

The data that support the findings of this study are available from the corresponding author upon reasonable request.

## Code availability

The MATLAB code used in this study is available (with restrictions) upon reasonable request.

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

## Acknowledgments

I thank Boris Galvan, Gunnar Jansen, Reza Sohrabi, Batoul Gisler for helpful discussions, and Thanushika Gunatilake for discussions and the ZMAP analysis.

## Author contributions

S.A.M. conceived and developed the idea, performed the simulations, and wrote the manuscript.

## Competing interests

The authors declare no competing interests.
