## [Peer Review File · Nature Communications]

Reviewers' comments:

Reviewer #1 (Remarks to the Author):

This is an interesting and provocative paper suggesting that Omori's law is the consequence of fluid pressure diffusion governed by the dynamical changes in permeability due to rupture-induced opening of fractures. Based on numerical experiments, the Author is able to demonstrate (1) that a dominant fraction of observed events is embedded in the high-pore fluid pressure area predicted by the numerical model and (2) that the observed Omori's law can be accurately reproduced by synthetic events based on a single fitting parameter ALPHA responsible for the time evolution (healing) of permeability after fracturing. The paper is strongly and (mostly) satisfactory revised in response to previous comments and concerns raised by reviewers. I noticed few minor issues that need further attention in term of (1) discussion of potential causes for fluid-absent environments, (2) numerical modeling procedure description and (3) numerical modeling results presentation. Specific comments are given below.

Taras Gerya, Zurich, 20.09.2019

Specific comments

Line 170. "With the exception of ALPHA, I used the same parameters in all simulations. As shown below, ALPHA determines the fit to the Omori-Utsu Law, and therefore constrained by the data." It would be important to discuss obtained best-fit ALPHA values representing characteristic timescales of fracture healing. Are these values similar for different events? Are they physically meaningful? It would also be important to describe fitting procedure in more details: is it manual or automated fitting? What fitting criteria are used? Showing and discussing (in a supplement) differences between the best-fit and worth-fit model examples will be very useful for better understanding physics of the process.

Figure 1. Caption "A lack of a volcanic arc suggests that flat subduction in Peru results in P-T paths that avoid important dehydration phase boundaries." Which phase boundaries do you mean? Serpentine dehydration? Modeling of slab PT paths for Peru (Siracuse et al., 2010, Fig. S1) suggest that they cross many dehydration boundaries at shallow depths. Arc absence would typically mean absence of fluid-fluxed melting in the mantle wedge that does not however preclude the presence of hydration in the forearc region. There can be some other reasons for the lack of the fluid above the subduction interface such as complex fluid paths driven by tectonic stresses in the overriding plate (e.g., Menant et al., 2019).

Line 56. "The lack of any volcanoes in Peru, nor detection of non-volcanic tremor (NVT), provides strong evidence that fluids are not being generated in significant quantities by the subduction process beneath Peru, and that the M8 earthquake occurred in dry lithosphere." Same as above. The statement about insignificant fluid generation in the slab under Peru is definitely too strong and oversimplified (or at least needs better support from subduction models, Siracuse et al., 2010, Fig. S1). The discussion of fluid presence above the subduction interface needs to be more nuanced.

Line 214. "It should be noted that there are other free parameters in the model (e.g. k_0 , d , s^*), but these were kept constant for all simulations and only ALPHA was varied and constrained by the data." It is important to show model sensitivity to variations in these other parameters (in a supplement). Otherwise it remains unclear how much conclusions are dependent on the model parameters choices.

Methods. It is important to describe the procedure of modeling more accurately. How are synthetic events detected and counted? Are results dependent on model resolution (in space and time)? Fitting procedure for ALPHA needs to be described. Supplementary table summarizing parameters and results of performed numerical experiments (with some supplementary figures) would also be very useful for understanding model sensitivity to various parameters.

Figures 3,4. As pointed out by reviewers, it would be important to compare results with best-fit models in a time progression. The pressurized fluid area growth outward and upward. Is this also the case for the aftershocks? Showing all events on the final pressure contours in Fig. 4 does not give full impression on space-time dynamics (some supplementary figures can help).

References

- Menant, A., Angiboust, S., Gerya, T. (2019) Stress-driven fluid flow controls long-term megathrust strength and deep accretionary dynamics. *Scientific Reports*, 9, Article Number: 9714
- Syracuse, E.M., van Keken, P.E., Abers, G.A. (2010) The global range of subduction zone thermal models. *Physics of the Earth and Planetary Interiors* 183, 73–90.

Reviewer #2 (Remarks to the Author):

This revised manuscript develops a model of aftershock generation through the creation of permeability and increased fluid pressure, and applies the model to a few mainshocks in southern California. The evidence that the proposed fluid model actually fits the observed aftershocks of the example sequences is still not very strong. Additionally, the motivation for the idea that aftershocks must be fluid driven is now justified by a comparison between upper-crustal earthquakes and intermediate-depth intraslab earthquakes, which is not a very compelling comparison.

One important point from the first reviews that wasn't addressed, and needs to be, is the spatial migration of the aftershocks. If the aftershocks are fluid-driven, we would expect to see the migration of earthquake locations as the fluid pressure diffuses. In the proposed model, the permeability evolves with time, so we may expect to see a more complicated diffusion of fluid pressure. It is not demonstrated in the manuscript, however, that the proposed model can correctly model the migration (or lack of migration) of the aftershock locations. All that's shown (Figure 4) is that the width of the fluid pressure zone at the end of the selected time period (the choice of which isn't explained, and that varies between mainshocks) is generally similar to the overall width of the seismicity in that time period. What's needed is a distance-versus-time plot showing the aftershocks and the modeled fluid pressure diffusion.

The other evidence presented for the fluid model is the fit to the aftershock decay curve (Figure 5). However, the number of aftershocks in this figure is normalized, which raises the question of whether the fluid model is capable of modeling the number of aftershocks, or whether that would need to be an additional adjustable parameter. Additionally, the first day of data is omitted from Figure 5. The early aftershock rate is typically flat for some period of time (usually less than a day), before the rate begins to decay. Because the first day is omitted, it's unclear what the fluid model predicts and whether it's compatible with this early behavior. The manuscript says that the first day is omitted because other physical mechanisms may also be triggering early aftershocks. However, we at least need to see that the fluid model alone doesn't overpredict the number of early aftershocks.

The comparison between the Peru and Mexico City earthquakes with no detected aftershocks, and the California earthquakes (Ridgecrest, Joshua Tree, Landers, Hector Mine, and Northridge) with many aftershocks is supposed to motivate this study by making the case that the occurrence of aftershocks depends on the presence of fluids. However, these particular earthquakes can't be used to differentiate between fluids and other possible controls on aftershock productivity. The Peru and Mexico City earthquakes are intermediate depth events occurring within the subducted oceanic slab, while the California events occur in the continental upper crust. It's well-known that intermediate depth earthquakes produce fewer aftershocks than shallow earthquakes (e.g. Frohlich, 1987). This isn't necessarily because of differences in fluid pressure, however, and could alternatively be due to other differences such as pressure, temperature, and/or mineralogy. A better motivation for this study would be to demonstrate a difference in aftershock rate between earthquakes in locations with different inferred fluid content, but otherwise in a similar tectonic setting. As it is, the statements that fluids are necessary to produce aftershocks are overstated.

Line 11. Change "no aftershocks" to "no aftershocks large enough to be detected by global seismic networks." I believe UNAM detected some small aftershocks following the Mexico earthquake.

Line 39. Note that not all statistical models assume Bath's Law, and can include differences in aftershock

productivity between regions, including very low productivity (e.g. Page et al., 2016).

Line 182. What data is used to determine the initial location of the over-pressured zone? This should be the location of the mainshock rupture, so a mainshock finite fault model is required.

Figure 2. How were the blue areas (used to compare with the model) chosen? They are hardly the only areas that could be projected in 2D for plotting.

Figure 5. The Omori A value isn't meaningful, and doesn't need to be reported, because the total number of events is normalized.

Reference:

Frohlich, C. (1987). Aftershocks and temporal clustering of deep earthquakes. *Journal of Geophysical Research: Solid Earth*, 92(B13), 13944-13956.

Page, M. T., Van Der Elst, N., Hardebeck, J., Felzer, K., & Michael, A. J. (2016). Three ingredients for improved global aftershock forecasts: Tectonic region, time-dependent catalog incompleteness, and intersequence variability. *Bulletin of the Seismological Society of America*, 106(5), 2290-2301.

Reviewer #3 (Remarks to the Author):

The author suggests that aftershock sequences are essentially the result of postseismic redistribution of overpressured fluids in the crust. As a concept I think it makes good sense, and the few examples provided appear to be consistent with this hypothesis. Having said that, the author could do better to refrain from overstating his case (e.g., especially in the abstract and conclusions). It's an interesting idea that is hardly proved by the simple models presented. I have doubts as to whether thermal pressurization would be capable of generating aftershock sequences that extend significantly beyond the fault plane because the thermal heating zone from a single earthquake is extremely narrow, far narrower (by a few orders of magnitude) than the initial width of overpressured fluids used in the models (as discussed more below). Given this, the author might consider an alternative scenario, such as the breaching of an overpressured reservoir at depth (which I believe might work better). All in all, it's a refreshing perspective and is sure to generate interest. Below are some detailed comments referring to specific points in the main text that might serve to improve the manuscript.

Line 12: '... contrary to all existing physical models of aftershock genesis that predict many thousands of aftershocks

First, I'm not sure I agree that all existing models predict aftershocks. Rate and state models don't normally produce aftershocks, though they produce afterslip which decays with 1/time. This is not really the same thing. Second, 'many thousands' is too loose.

Line 16: 'Here I show that the Peru...' The author doesn't really show it, he proposes it.

Line 19: Is necessary to generate, (or favors)?

Line 20: All! This might be considered an overstatement.

Line 21: To test what? (whether they are all driven by fluid?). How would a model show that?

Line 37: Is this statement really true? (the current earthquake hypothesis doesn't really concern aftershocks, in fact it really doesn't directly explain them).

Line 40: Then the hypothesis doesn't really fail there. But the question is valid, why do some earthquakes have more pronounced aftershock sequences than others? Maybe it should be posed like that.

Line 145: As written, eqn 4 is not a nonlinear diffusion equation. But it's true that introducing $k=f(P)$ makes it nonlinear. A little more care could be taken.

Lines 165-167: The models are based on an initially overpressured fault zone. While this could be expected from thermal pressurization, I doubt whether the thickness of the overpressured zone could reach values

used in the model (I didn't find what the initial width was, but it seems to be on the order of 500m, judging from Fig 3). Although faults might experience coseismic heating (and pressurization), thermal models show that this occurs over a very narrow width (far less than 1 m in width). This hot narrow zone then decays extremely fast (in less than 1 day, by conduction). The rocks a few meters away will see almost nothing from a single earthquake, even if the fault reaches melting temperature (of course there is a cumulative heating over multiple cycles). These arguments show that the zone of pressurization due to shear heating should be quite narrow, far narrower than 500m. As the initial zone width of overpressured fluids is made narrower, I expect there won't be enough overpressured fluid to extend far beyond the fault following rupture.

Line 181: But what about its width?

Line 227: Yes, I suspect this might be the strongest argument, based on the above comments concerning the width of a thermally pressurized zone.

Line 279: Theta has a precise definition, which should be given.

Reviewers' comments:

Reviewer #1 (Remarks to the Author):

This is an interesting and provocative paper suggesting that Omori's law is the consequence of fluid pressure diffusion governed by the dynamical changes in permeability due to rupture-induced opening of fractures. Based on numerical experiments, the Author is able to demonstrate (1) that a dominant fraction of observed events is embedded in the high-pore fluid pressure area predicted by the numerical model and (2) that the observed Omori's law can be accurately reproduced by synthetic events based on a single fitting parameter ALPHA responsible for the time evolution (healing) of permeability after fracturing. The paper is strongly and (mostly) satisfactory revised in response to previous comments and concerns raised by reviewers. I noticed few minor issues that need further attention in term of (1) discussion of potential causes for fluid-absent environments, (2) numerical modeling procedure description and (3) numerical modeling results presentation. Specific comments are given below.
Taras Gerya, Zurich, 20.09.2019

I thank Prof. Gerya for these comments.

Specific comments

Line 170. "With the exception of ALPHA, I used the same parameters in all simulations. As shown below, ALPHA determines the fit to the Omori-Utsu Law, and therefore constrained by the data." It would be important to discuss obtained best-fit ALPHA values representing characteristic timescales of fracture healing. Are these values similar for different events? Are they physically meaningful?

I appreciate this very insightful comment because it exposed what is really going on in the model and from which I gained fundamental insight.

Figure 6 in the new manuscript shows that the values for α are physically meaningful, with characteristic timescales, and that I no longer adjust α to fit Omori-Utsu, but rather let the data and the p-value (inset Figure 1) determine the likely value for α before running the simulation.

In this new manuscript, I no longer compare model, Omori, and data in the same plot because the model fits the data better than the Omori-Utsu Law. I now separate the ZMAP fitting of the data to Omori in different time windows from the model fitting of the same data in the same time windows. The model fits for all 10 simulated aftershock sequences are excellent in all time windows.

It would also be important to describe fitting procedure in more details: is it manual or automated fitting? What fitting criteria are used? Showing and discussing (in a supplement) differences between the best-fit and worth-fit model examples will be very useful for better understanding physics of the process.

This is another insightful comment. Everything was manual in the previous manuscript, and I was using c-values of 8-12 days because that is what to my eye best fit the data even though reported c-values are less than a day. This is completely changed in the new manuscript.

In this new manuscript, the p- and c-values were systematically determined from the data using the ZMAP statistical bootstrapping algorithm. With this method, I find that the p- and c-values depend on the time window of interest, with rather poor fits to the data in most cases. I consistently followed the same procedure and so other researchers can reproduce these results.

Model fitting to the data is not statistical, but the visual comparisons to the data are compelling. With Figure 6 of the new manuscript in hand, I can now determine the best-fitting ALPHA (before starting a simulation) by the statistical determination of the p-value. One main point of this new manuscript is that Omori-Utsu is actually not very good at fitting the data, while all modelling (Figures 3,4,5, S4, S5, and S6) show excellent fits to the same data.

Figure 1. Caption “A lack of a volcanic arc suggests that flat subduction in Peru results in P-T paths that avoid important dehydration phase boundaries.” Which phase boundaries do you mean? Serpentine dehydration?

Note: This comment and the ones that follow relate to the geodynamics and petrology of Peru/Mexico. The discussion about this in the old manuscript has now been shifted to Supplemental because (although relevant), it now plays only a supporting role of a larger discussion about the wide spectrum of aftershock behaviors in this new manuscript. But each of these comments are addressed in Supplemental.

I had in mind Serpentine dehydration because most (all?) volcanic arcs are driven by this phase transition. This is now clarified in the discussion of Peru and Mexico now found in Supplemental.

Modeling of slab PT paths for Peru (Siracuse et al., 2010, Fig. S1) suggest that they cross many dehydration boundaries at shallow depths.

I agree that with this statement, and these dehydration boundaries likely coincide with the vast ore deposits in the fore-arc. But these normal faulting earthquakes occurred between these two regions, just as they did in the Mexico earthquake case.

Arc absence would typically mean absence of fluid-fluxed melting in the mantle wedge that does not however preclude the presence of hydration in the forearc region. There can be some other reasons for the lack of the fluid above the subduction interface such as complex fluid paths driven by tectonic stresses in the overriding plate (e.g., Menant et al., 2019).

I agree with the sentiment of these comments and this reference has been included in Supplemental.

Line 56. “The lack of any volcanoes in Peru, nor detection of non-volcanic tremor (NVT), provides strong evidence that fluids are not being generated in significant quantities by the subduction process beneath Peru, and that the M8 earthquake occurred in dry lithosphere.” Same as above.

My response is the same as above.

The statement about insignificant fluid generation in the slab under Peru is definitely too strong and oversimplified (or at least needs better support from subduction models, Syracuse et al., 2010, Fig. S1). The discussion of fluid presence above the subduction interface needs to be more nuanced.

This is now very subdued in the Introduction, and is only part of a larger introduction of a wide spectrum of aftershock behaviors.

Line 214. “It should be noted that there are other free parameters in the model (e.g. k_0 , d , s^*), but these were kept constant for all simulations and only ALPHA was varied and constrained by the data.” It is important to show model sensitivity to variations in these other parameters (in a supplement). Otherwise it remains unclear how much conclusions are dependent on the model parameters choices.

This is an important point, and has been addressed by example in the new manuscript. Namely, σ_{star} and k_0 were chosen to create a background permeability field that approximates the Ingebritsen/Manning model of crustal permeability, and a three order of magnitude increase in permeability seems to me reasonable.

The example in the new manuscript is that Northridge and the Stepover are both well-matched, even though ALPHA plays no role in the Stepover case (Figure 3) over the timescale of the simulation. This indicates that σ_{star} , d , and k_0 are relatively limited in range, and were therefore retained the same values for all simulations. Future experimental studies could further constrain these parameters.

I have now included permeability and pressure-time histories for the simulations because it exposes the onset of alpha and the consequent shutdown of permeability.

Methods. It is important to describe the procedure of modeling more accurately. How are synthetic events detected and counted?

Everything about the numerical model is described in detail in Miller, Geofluids (2015), where the model was calibrated against a meso-scale fluid-injection experiment in Basel Switzerland. Nevertheless, I have expanded (in Methods and Supplemental) the modelling procedure in more detail.

The only differences with the Basel simulation are (1) the pressure-time history driving the earthquakes was prescribed in Basel because it was measured, and (2) ALPHA was not

included because I had not yet developed it, but has no influence because I only simulated about 10 days of Basel.

Are results dependent on model resolution (in space and time)? Fitting procedure for ALPHA needs to be described. Supplementary table summarizing parameters and results of performed numerical experiments (with some supplementary figures) would also be very useful for understanding model sensitivity to various parameters.

Simulation results using 200x200 numerical grid points are indistinguishable from simulation results on 300x300 numerical grid points. For time-stepping, I began simulations with a very small timestep, but then progressively increased it as pressure gradients reduced because each simulation took about 30 hours. I noticed no change in overall behavior using various time-stepping schemes, so although I cannot say that there is absolutely no effect, if there is, then it not significant.

Figures 3,4. As pointed out by reviewers, it would be important to compare results with best-fit models in a time progression. The pressurized fluid area growth outward and upward. Is this also the case for the aftershocks? Showing all events on the final pressure contours in Fig. 4 does not give full impression on space-time dynamics (some supplementary figures can help).

I now include a Supplemental Figure (S3) showing typical x-t and z-t model results compared with observations. The model does show a diffusion front in the x-direction (as expected), but the data is very messy and I doubt any diffusion front could be picked up with the seismic network installed at the time.

Importantly, recent studies of Ridgecrest aftershocks¹ do show migration, so more precise data in the future could reveal significant migration.

References

- Menant, A., Angiboust, S., Gerya, T. (2019) Stress-driven fluid flow controls long-term megathrust strength and deep accretionary dynamics. *Scientific Reports*, 9, Article Number: 9714
- Syracuse, E.M., van Keken, P.E., Abers, G.A. (2010) The global range of subduction zone thermal models. *Physics of the Earth and Planetary Interiors* 183, 73–90.

Reviewer #2 (Remarks to the Author):

This revised manuscript develops a model of aftershock generation through the creation of permeability and increased fluid pressure, and applies the model to a few mainshocks in southern California.

The evidence that the proposed fluid model actually fits the observed aftershocks of the example sequences is still not very strong.

The new manuscript, I think, disputes this comment. Comparisons between modelling and observations are now quite compelling. The problem in the previous manuscript is that by plotting data, model, and Omori on the same looked somewhat messy. I now separate data comparisons with Omori-Utsu from data comparisons with model results because the model fits the data better than Omori-Utsu.

Additionally, the motivation for the idea that aftershocks must be fluid driven is now justified by a comparison between upper-crustal earthquakes and intermediate-depth intraslab earthquakes, which is not a very compelling comparison.

This is now addressed in the Introduction of the new manuscript, where I bring in more observations to support my case and summarized in Supplemental Table 1. The discussion about the Peru and Mexico earthquakes is now relegated to Supplemental because this is now only one part of a larger discussion about aftershock behaviors discussed in the Introduction.

One important point from the first reviews that wasn't addressed, and needs to be, is the spatial migration of the aftershocks. If the aftershocks are fluid-driven, we would expect to see the migration of earthquake locations as the fluid pressure diffuses. In the proposed model, the permeability evolves with time, so we may expect to see a more complicated diffusion of fluid pressure. It is not demonstrated in the manuscript, however, that the proposed model can correctly model the migration (or lack of migration) of the aftershock locations. All that's shown (Figure 4) is that the width of the fluid pressure zone at the end of the selected time period (the choice of which isn't explained, and that varies between mainshocks) is generally similar to the overall width of the seismicity in that time period. What's needed is a distance-versus-time plot showing the aftershocks and the modeled fluid pressure diffusion.

See previous explanation from Reviewer 1. I have now included x-t and z-t plots (Figure S3), with a discussion about why they do not match well the observations.

The other evidence presented for the fluid model is the fit to the aftershock decay curve (Figure 5). However, the number of aftershocks in this figure is normalized, which raises the question of whether the fluid model is capable of modeling the number of aftershocks, or whether that would need to be an additional adjustable parameter.

I find it reasonable to normalize by the maximum number of observed and modeled events. “Aftershocks” in the model are simply numerical grid points that reach the Mohr-Coulomb failure condition (Figure S2), so comparing the absolute number of failing numerical grid points with real aftershocks is not reasonable because the number of aftershocks would definitely be grid-size dependent (e.g. double the grid points, double the number of numerical aftershocks). Given sufficient resolution, the overall results are not grid-size dependent.

I have included additional Figures (S2) to illustrate how numerical aftershocks are generated and counted, and I think this may clear up the reviewer’s concern.

Additionally, the first day of data is omitted from Figure 5. The early aftershock rate is typically flat for some period of time (usually less than a day), before the rate begins to decay. Because the first day is omitted, it’s unclear what the fluid model predicts and whether it’s compatible with this early behavior. The manuscript says that the first day is omitted because other physical mechanisms may also be triggering early aftershocks. However, we at least need to see that the fluid model alone doesn’t overpredict the number of early aftershocks.

This is related to the c -value, which is the time from the earthquake to Omori-Utsu decay. As pointed out by the reviewer, this is less than a day, but no $c < 1$ day fits any of the modelled sequences. But I think more importantly, I do not want to overplay this model by trying to claim every aftershock, from $t=0$ to $t=1000$ days is fluid-driven. This is especially true because like Kunlun, Jamaica, and Big Bear, numerous processes are in play immediately after an earthquake, including dynamic shaking, dynamic and static stress changes, and an overall readjustment of the system.

In addition, the high-pressure region in the model, when initiated at $t=0$, results in failure of all grid-points in the immediate neighborhood, which results (in my opinion) in an artefact of the initial conditions. Fluid flow may very well be the reason for a change from a flat aftershock rate to before the onset of a decay because the system changes from a stress-dominated (fracture creation) to fluid-dominated (fluid pressure diffusion), but addressing this is beyond the scope of the current paper.

It is also interesting that the Stepover showed constant aftershock rates lasting about 2 years, which is intriguing and could indicate a rich fluid source at depth. A swarm of earthquakes triggered by the Hector Mine earthquake (Figure S4) is also intriguing since triggered seismicity is generally thought to be limited to hydrothermal regions.

The comparison between the Peru and Mexico City earthquakes with no detected aftershocks, and the California earthquakes (Ridgecrest, Joshua Tree, Landers, Hector Mine, and Northridge) with many aftershocks is supposed to motivate this study by making the case that the occurrence of aftershocks depends on the presence of fluids. However, these particular earthquakes can’t be used to differentiate between fluids and other possible controls on aftershock productivity. The Peru and Mexico City earthquakes are intermediate depth events occurring within the subducted oceanic slab, while the California events occur in the continental upper crust. It’s well-known that intermediate depth earthquakes produce fewer aftershocks than shallow earthquakes (e.g. Frohlich, 1987). This isn’t necessarily because of differences in fluid pressure, however, and could alternatively be due to other differences such as pressure, temperature, and/or mineralogy.

A better motivation for this study would be to demonstrate a difference in aftershock rate between earthquakes in locations with different inferred fluid content, but otherwise in a similar tectonic setting. As it is, the statements that fluids are necessary to produce aftershocks are overstated.

I agree with this comment, and have included additional discussion in the Introduction, caveats, and Table (S1) that summarizes observations.

Line 11. Change “no aftershocks” to “no aftershocks large enough to be detected by global seismic networks.” I believe UNAM detected some small aftershocks following the Mexico earthquake.

This has been included.

Line 39. Note that not all statistical models assume Bath’s Law, and can include differences in aftershock productivity between regions, including very low productivity (e.g. Page et al., 2016).

Bath’s Law is no longer discussed.

Line 182. What data is used to determine the initial location of the over-pressured zone? This should be the location of the mainshock rupture, so a mainshock finite fault model is required.

I impose the initial location of the over-pressured zone at the center of the profiles (identified by the yellow lines in Figure 1). The fault extends out of the page, so finite fault model is not useful.

Figure 2. How were the blue areas (used to compare with the model) chosen? They are hardly the only areas that could be projected in 2D for plotting.

Figure 1 shows all of the sequences studied so far. I am confident that all red dots could turn blue in future studies.

Figure 5. The Omori A value isn’t meaningful, and doesn’t need to be reported, because the total number of events is normalized.

This is included now for Equation (1).

Reference:

Frohlich, C. (1987). Aftershocks and temporal clustering of deep earthquakes. *Journal of Geophysical Research: Solid Earth*, 92(B13), 13944-13956.

Page, M. T., Van Der Elst, N., Hardebeck, J., Felzer, K., & Michael, A. J. (2016). Three ingredients for improved global aftershock forecasts: Tectonic region, time-dependent catalog incompleteness, and intersequence variability. *Bulletin of the Seismological Society*

of America, 106(5), 2290-2301.

Reviewer #3 (Remarks to the Author):

The author suggests that aftershock sequences are essentially the result of postseismic redistribution of overpressured fluids in the crust. As a concept I think it makes good sense, and the few examples provided appear to be consistent with this hypothesis. Having daid that, the author could do better to refrain from overstating his case (e.g., especially in the abstract and conclusions). It's an interesting idea that is hardly proved by the simple models presented. I have doubts as to whether thermal pressurization would be capable of generating aftershock sequences that extend significantly beyond the fault plane because the thermal heating zone from a single earthquake is extremely narrow, far narrower (by a few orders of magnitude) than the initial width of overpressured fluids used in the models (as discussed more below). Given this, the author might consider an alternative scenario, such as the breaching of a overpressured reservoir at depth (which I believe might work better). All in all, it's a refreshing perspective and is sure to generate interest. Below are some detailed comments referring to specific points in the main text that might serve to improve the manuscript.

I appreciate these comments, and all overstatements are subdued (or supported) in the new manuscript. Very little of the previous manuscript remains in this new manuscript.

Line 12: '... contrary to all existing physical models of aftershock genesis that predict many thousands of aftershocks

First, I'm not sure I agree that all existing models predict aftershocks. Rate and state models don't normally produce aftershocks, though they produce afterslip which decays with 1/time. This is not really the same thing. Second, 'many thousands' if too loose.

This sentence does not appear in the new manuscript.

Line 16: 'Here I show that the Peru...' The author doesn't really show it, he proposes it.

The Peru earthquake is now just a small aspect of the larger picture, and does not appear in the abstract.

Line 19: Is necessary to generate, (or favors)?

This has been modified.

Line 20: All! This might be considered an overstatement.

This has been removed

Lien 21: To test what? (whether they are all driven by fluid?). How would a model show that?

This has been removed in the new manuscript.

Line 37: Is this statement really true? (the current earthquake hypothesis doesn't really concern aftershocks, in fact it really doesn't directly explain them).

This has been removed in the new manuscript.

Line 40: Then the hypothesis doesn't really fail there. But the question is valid, why do some earthquake have more pronounced aftershock sequences than others? Maybe it should be posed like that.

I appreciate this comment because the new manuscript has been reformatted to pose the question just like that... "Why do some earthquakes have more pronounced aftershock sequences than others?"

Line 145: As written, eqn 4 is not a nonlinear diffusion equation. But it's true that introducing $k=f(P)$ makes it nonlinear. A little more care could be taken.

This has been modified

Lines 165-167: The models are based on an initially overpressured fault zone. While this could be expected from thermal pressurization, I doubt whether the thickness of the overpressured zone could reach values used in the model (I didn't find what the initial width was, but it seems to be on the order of 500m, judging from Fig 3). Although faults might experience coseismic heating (and pressurization), thermal models show that this occurs over a very narrow width (far less than 1 m in width). This hot narrow zone then decays extremely fast (in less than 1 day, by conduction). The rocks a few meters away will see almost nothing from a single earthquake, even if the fault reaches melting temperature (of course there is a cumulative heating over multiple cycles). These arguments show that the zone of pressurization due to shear heating should be quite narrow, far narrower than 500m. As the initial zone width of overpressured fluids is made narrower, I expect there won't be enough overpressured fluid to extend far beyond the fault following rupture.

I agree that thermal pressurization, as its typically discussed, is limited in width. I now discuss this in the new manuscript, and now emphasize devolatilization processes and linking to suprahydrostatic reservoirs at depth.

Line 181: But what about its width?

In all cases, I selected all aftershocks that appeared to belong to one group. It is possible that a few events missed out, but I grabbed all (thousands) that appeared related to a particular section (named now in Figure 1).

Line 227: Yes, I suspect this might be the strongest argument, based on the above comments concerning the width of a thermally pressurized zone.

This is now emphasized because I agree that expansion of pore fluids is limited in volume and would not be able to drive the system for very long.

Line 279: Theta has a precise definition, which should be given.

This is now given

- 1 Shelly, D. R. A High-Resolution Seismic Catalog for the Initial 2019 Ridgecrest Earthquake Sequence: Foreshocks, Aftershocks, and Faulting Complexity. *Seismological Research Letters* (2020).

REVIEWERS' COMMENTS

Reviewer #1 (Remarks to the Author):

The paper has been critically improved by the major revision that carefully addressed previous concerns. In support of the proposed fault pressurization model, I would like to mention that coseismic self-pressurization of fluid-bearing fault zones can also be attributed to elasto-plastic collapse of rapidly deforming fluid-filled pores – the new pressurization mechanism that has recently been identified on the basis of fully coupled poro-elasto-visco-plastic seismo-hydro-mechanical model (Petrini et al., 2020).

Taras Gerya, Zurich,

Petrini, C., Gerya, T., Yarushina, V., van Dinther, Y., Connolly, J., Madonna, C. (2020) Seismo-hydro-mechanical modelling of the seismic cycle: Methodology and implications for subduction zone seismicity, *Tectonophysics*, 791, 228504.

Reviewer #3 (Remarks to the Author):

The author appears to have made a good attempt at improving the manuscript. I am favorable for publication with very little modification. I simply suggest that the text in the various panels of the figures could be made uniform, since a variety of fonts are currently used. Also, on line 48, I suggest using 'differences' rather than 'discrepancies'.

Response to referees:

The paper has been critically improved by the major revision that carefully addressed previous concerns. In support of the proposed fault pressurization model, I would like to mention that coseismic self-pressurization of fluid-bearing fault zones can also be attributed to elasto-plastic collapse of rapidly deforming fluid-filled pores – the new pressurization mechanism that has recently been identified on the basis of fully coupled poro-elasto-visco-plastic seismo-hydro-mechanical model (Petrini et al., 2020).

Taras Gerya, Zurich,

This reference is now included, line 164.

Reviewer #3 (Remarks to the Author):

The author appears to have made a good attempt at improving the manuscript. I am favorable for publication with very little modification. I simply suggest that the text in the various panels of the figures could be made uniform, since a variety of fonts are currently used. Also, on line 48, I suggest using 'differences' rather than 'discrepancies'.

This has been changed as requested.